# Transcription Factor MAFB as a Prognostic Biomarker for the Lung Adenocarcinoma

**DOI:** 10.3390/ijms23179945

**Published:** 2022-09-01

**Authors:** Omar Samir, Naohiro Kobayashi, Teppei Nishino, Mennatullah Siyam, Manoj Kumar Yadav, Yuri Inoue, Satoru Takahashi, Michito Hamada

**Affiliations:** 1Laboratory Animal Resource Center in Transborder Medical Research Center, Faculty of Medicine, University of Tsukuba, 1-1-1 Tennodai, Tsukuba 305-8575, Japan; 2Department of Pathology, Faculty of Veterinary Medicine, Mansoura University, Mansoura 35516, Egypt; 3Department of General Thoracic Surgery, Faculty of Medicine, University of Tsukuba, 1-1-1 Tennodai, Tsukuba 305-8575, Japan; 4Department of Anatomy and Embryology, Faculty of Medicine, University of Tsukuba, 1-1-1 Tennodai, Tsukuba 305-8575, Japan; 5Department of Medical Education and Training, Tsukuba Medical Center Hospital, 1-3-1 Amakubo, Tsukuba 305-8558, Japan; 6Ph.D. Program in Human Biology, School of Integrative and Global Majors, University of Tsukuba, 1-1-1 Tennodai, Tsukuba 305-8575, Japan; 7Doctoral Program in Biomedical Sciences, Graduate School of Comprehensive Human Sciences, University of Tsukuba, 1-1-1 Tennodai, Tsukuba 305-8575, Japan; 8International Institute for Integrative Sleep Medicine (WPI-IIIS), University of Tsukuba, 1-1-1 Tennodai, Tsukuba 305-8575, Japan; 9Life Science Center for Survival Dynamics, Tsukuba Advanced Research Alliance (TARA), University of Tsukuba, 1-1-1 Tennodai, Tsukuba 305-8577, Japan

**Keywords:** biomarker, cancer severity, cancer prognosis, MAFB, tumor-associated macrophages

## Abstract

MAFB is a basic leucine zipper (bZIP) transcription factor specifically expressed in macrophages. We have previously identified MAFB as a candidate marker for tumor-associated macrophages (TAMs) in human and mouse models. Here, we analyzed single-cell sequencing data of patients with lung adenocarcinoma obtained from the GEO database (GSE131907). Analyzed data showed that general macrophage marker CD68 and macrophage scavenger receptor 1 (CD204) were expressed in TAM and lung tissue macrophage clusters, while transcription factor MAFB was expressed specifically in TAM clusters. Clinical records of 120 patients with lung adenocarcinoma stage I (*n* = 57), II (*n* = 21), and III (*n* = 42) were retrieved from Tsukuba Human Tissue Biobank Center (THB) in the University of Tsukuba Hospital, Japan. Tumor tissues from these patients were extracted and stained with anti-human MAFB antibody, and then MAFB-positive cells relative to the tissue area (MAFB^+^ cells/tissue area) were morphometrically quantified. Our results indicated that higher numbers of MAFB^+^ cells significantly correlated to increased local lymph node metastasis (nodal involvement), high recurrence rate, poor pathological stage, increased lymphatic permeation, higher vascular invasion, and pleural infiltration. Moreover, increased amounts of MAFB^+^ cells were related to poor overall survival and disease-free survival, especially in smokers. These data indicate that MAFB may be a suitable prognostic biomarker for smoker lung cancer patients.

## 1. Introduction

V-maf musculoaponeurotic fibrosarcoma oncogene homolog B (MAFB) belongs to the large Maf transcription factor family and is a bzip transcription factor that regulates target gene expression [1]. *Mafb* is expressed in several tissues and is associated with the differentiation of various cell types, such as kidney podocytes [2], keratinocytes [3], and pancreatic α-cells and β-cells [3,4].

In the hematopoietic cell lineage, a transcriptome analysis using multi-dendritic cell (DC) and macrophage subsets showed that the expression of *Mafb* is associated specifically with monocyte-macrophage lineage and not DC lineage [5]. An increase in *Mafb* expression was observed in anti-inflammatory M2-type macrophages in vitro [6]. Moreover, MAFB in macrophages plays an essential role in resolving inflammation in ischemic conditions, efferocytosis preventing autoimmunity, and inhibiting macrophage apoptosis in atherogenic conditions [7,8,9], indicating that MAFB regulates the homeostatic function of macrophages. Lung alveolar macrophages (AM) express a low level of MAFB [5]; however, exposure to cigarette smoke progressively increased the expression of *Mafb* in a mouse model [10]. Even though patients with chronic obstructive pulmonary disease also exhibit increased *Mafb* expression [11], the relationship between exposure to cigarette smoke, *Mafb* expression, and lung cancer remains largely unidentified.

Tumor-associated macrophages (TAM) are the major cell populations of the tumor microenvironment (TME) and promote tumor progression, metastasis, angiogenesis, and resistance to therapy [12]. A higher infiltration of TAMs is often associated with a high mortality rate in various cancers [13]. M2 macrophage markers such as CD163, CD68, CD206, and CD204 are TAM markers that are widely used to assess cancer progression [14]. However, distinguishing M1 and M2 macrophages within the TAM in vivo remains challenging [15]. Consistent with this, CD163 and CD206 are reported to be expressed on M1-like TAMs or DCs that stimulate T cell activity in gastrointestinal tumors and ovarian ascites [16,17]. CD204 is also expressed in dendritic cells [18] in angioblastic T cell lymphoma (AITL); however, CD204 was not expressed in TAM [19]. Even though CD68, CD163, and CD204 have been widely used to assess the severity and outcome of human cancers [14,20], opinion on what constitutes the definitive TAM marker remains controversial.

We have previously reported *Mafb* expression in M2-type TAMs in a mouse tumor model of Lewis lung carcinoma. Furthermore, we have shown a significant upregulation of *MAFB* in human lung carcinomas (stage I and stage III) [21]. However, the lack of expression of *Mafb* in AM, along with its cigarette smoke-induced increase in expression [5,10], led us to hypothesize that MAFB could be a potential TAM marker for lung cancer. Here, we analyzed the single-cell RNA sequence (scRNA-seq) data of human lung cancer previously reported to exhibit a lack of *MAFB* expression in AM but expressed specifically in another macrophage lineage. Further, the lung tissue of 120 patients with lung cancer was immunostained using an anti-MAFB antibody, and the association between MAFB and cancer-related parameters was analyzed. Our results indicated that *Mafb* is highly specific for TAM and is a potential prognostic marker. Moreover, MAFB was also identified as a prognostic marker that can predict the risk of mortality among smoker patients.

## 2. Results

### 2.1. MAFB Is Specifically Expressed in Monocytes/Macrophages but Not Alveolar Macrophages in Both Normal and Cancerous Tissue

It has been shown that in a mouse model, *Mafb* is not expressed in AM [5]. Expecting the same for humans, *MAFB* may be a more specific TAM marker in lung cancer than other M2 macrophage markers. Therefore, we compared the distribution of *MAFB* and other macrophage markers, *CD68* and *CD204*, using scRNA sequencing data of lung cancer patients, including normal lung, tumor tissue (stage I and III, *n* = 7), and advanced tumor tissue (stage IV, *n* = 4), as reported by Kim et al. (GSE131907) [22]. There were 34 clusters in all samples (Appendix A). The myeloid series was extracted and analyzed using myeloid markers, *LYZ*, *MARCO*, *CD68*, and *FCGR3A* (Appendix A). The extracted myeloid population had 18 (labeled 0–17) clusters, which were classified according to the expression of marker genes into AM, (cluster 0, 4, 10, 15), ML (cluster 2, 3, 5, 6, 9, 11, 13, 16), Mo (cluster 1, 8), and DC (cluster 7, 12, 14, 17) (Figure 1A and Appendix A). Interestingly, all macrophage populations were identified in normal lung, tumor, and advanced tumor samples. AM was found in cancer tissue; however, the number of AM decreased with advancement in cancer stage (Figure 1B).

We further analyzed the expression patterns of *CD68* and *CD204* in normal lung and cancer tissues. Our results showed that *CD68* was expressed in ML and Mo clusters in normal lung and cancer tissues; however, strongly expressed in AM clusters (Figure 1C,D). *CD204* was also expressed in the ML clusters in tumor tissues and the AM clusters (Figure 1E,F). Compared to *CD68* and *CD204*, *MAFB* was markedly less expressed in AM and specifically expressed in ML and Mo in all data from normal lung and cancer tissues (Figure 1G,H). A heatmap analysis confirmed *MAFB* expression in myeloid clusters, and the results showed lower *MAFB* expression in all the AM clusters (0, 4, 10, 15) compared with expression levels of *CD68* and *CD204* (Figure 1I). A higher level of *MAFB* expression was observed in the Mo cluster, than *CD68* and *CD204*, suggesting that *MAFB* is expressed in infiltrating monocytes (Figure 1J). Consistently, the expression pattern of CCR2, a chemokine receptor for monocytes, was similar to that of MAFB in the monocyte clusters (Appendix A). In the ML, *MAFB*, *CD68*, and *CD204* were all expressed in clusters 2, 5, and 6 of the stage I samples, while cluster 9 showed strong expression of *MAFB*. As for cluster 13, *CD68* was strongly expressed in stage I and *CD204* in the advanced tumor, but *MAFB* was not expressed (Figure 1K). These results indicate that *MAFB*, *CD68*, and *CD204* could be identified as markers of TAM but have different expression patterns among subsets of human macrophages.

### 2.2. Higher MAFB^+^ Cell Density May Be Associated with Poor Clinical Prognosis among Lung Cancer Patients

Cancer diagnostic TAM markers, *CD68*, *CD204*, *CD206*, and *CD163*, are expressed on lung tissue AM (Figure 1 and Appendix A); however, the poor *MAFB* expression on AM might significantly impact the assessment of cancer progression using TAM as an indicator. Therefore, to investigate whether the density of MAFB-positive cells is related to the clinical features of the tumor, we collected and analyzed cancer tissues from 120 patients with lung adenocarcinoma with or without nodal involvement (stage I, II, and III) admitted to the Tsukuba University Hospital between 2010 and 2019 (Appendix A). The cancer tissues were immunostained using an anti-MAFB antibody, and the number of MAFB-positive cells relative to the tissue area was counted. The patients were ranked according to the MAFB^+^ cell density into low (25%) (low-MAFB^+^ group; *n* = 30, MAFB^+^ density ≤ 0.005), mid (49%) (mid MAFB^+^ group; *n* = 59, MAFB^+^ density = 0.006–0.016), and the higher (26%) (high-MAFB^+^ group; *n* = 31, MAFB^+^ density ≥ 0.017). The signals of MAFB staining differed significantly in each group (Figure 2A,B). Moreover, the high-MAFB^+^ group presented with significantly large tumors (Figure 2C). The correlation between the three groups, low-, mid-, and high- MAFB^+^ cell density with the recorded clinical features of the patients was analyzed using Fisher’s exact test (Table 1).

The patients did not differ significantly in age < 70 years versus ≥ 70 years (*p* = 0.22), or smoking status (*p*-value = 0.30). The female-to-male ratios were significantly different in the mid-MAFB^+^ group (male: female, 21:38) and the high-MAFB^+^ group (male:female, 20:11, *p* < 0.03). No significant association between smoking habits and the MAFB cell population was identified; however, patients with smoking habits tended to cluster more in the high-MAFB^+^ group (never: former/current, 8:23). Most of the patients with stage I adenocarcinoma were grouped into the low-MAFB^+^ (stage I: stage III, 24:6), while a significant number of patients with stage III adenocarcinoma were grouped into the high-MAFB^+^ group (stage I:stage III, 3:28, *p* < 0.01). Similarly, clinical characteristics related to cancer recurrence, including nodal involvement, lymphatic permeation, and vessel invasion, were lower in tissues with low-MAFB^+^, while the high-MAFB^+^ group showed a significant correlation. Most of the tissues with low-MAFB^+^ showed no pleural infiltration. These findings suggest that higher MAFB^+^ cell density may be associated with poor clinical prognosis among patients with stages I, II, and III lung adenocarcinoma.

### 2.3. High-MAFB^+^ Cell Density Indicated a Higher Mortality Risk

Many studies have shown that TAM markers are associated with survival in lung cancer patients [13]. Therefore, we analyzed whether the MAFB^+^ cell density was associated with survival rates of the patients with non-metastatic (Stage I to III) lung adenocarcinoma. Figure 3A,B show the OS and DFS of low- (black line), mid- (green line), and high- (red line) MAFB^+^ cell density groups. The curves indicate that high-MAFB^+^ cell density indicated a higher mortality risk as the mean survival time (MST) for the low-, mid-, and high-MAFB^+^ groups were 114.7 months, 104.4 months, and 75.5 months (*p* < 0.001, log-rank test), respectively (Figure 3A).

In terms of DFS, the MST in the low-, mid-, and high-MAFB^+^ group was 96.4 months, 78.6 months, and 39.9 months (*p* < 0.001, log-rank test), respectively (Figure 3B). Compared with the MST of nodal involvement; negative (−) with an OS of 109.0 months and positive (+) with an OS of 92.4 months; the low-MAFB^+^ patients showed longer OS than patients with (−) nodal involvement and high-MAFB^+^ patients showed a 17 month shorter OS than patients with nodal involvement (Appendix A and Figure 3A,B).

Furthermore, Pearson correlation analysis showed that MAFB^+^ cell density was negatively correlated with OS or DFS (Figure 3C, R score: −0.37, *p* < 0.001, D, R score: −0.38, *p* < 0.001). Consistently, the univariate analysis using the Cox hazard test disclosed that the OS (low vs. Mid, *p* = 0.0791, low vs. high, *p* = 0.0011) and DFS (low vs. mid, *p* = 0.0828, low vs. high, *p* = 0.0018) was associated with MAFB expression and other factors except for sex and age (Table 2).

On the other hand, multivariate analysis suggested that MAFB expression was less influential than smoking history and nodal involvement. (Table 2).

These results suggest that grouping by the degree of MAFB expression allows a more detailed examination of hazard and mortality risk in patients with stage I to stage III lung adenocarcinoma. Thus, MAFB^+^ cell density may be an ideal predictor of the hazard ratio and DFS in these patients after surgery.

### 2.4. MAFB Could Be a Prognostic TAM Marker for Patients with Smoking Habits with Lung Adenocarcinoma

Univariate and multivariate analyses have revealed that smoking history and MAFB expression affect survival (Table 2). A previous study showed that cigarette smoke induces MAFB expression in lung macrophages in a mouse model [10]. Therefore, we decided to analyze whether there is any relationship between smoking and MAFB expression. We first checked the correlation between MAFB expression level and smoking index, but no association was observed (Appendix A). Next, we decided to ascertain whether the intensity of MAFB expression is related to survival in smokers and non-smokers. Our results showed that the OS rates of smokers (*n* = 75) were significantly lower compared to non-smokers (*n* = 45) (Appendix A). Furthermore, we examined OS and DFS of low, mid, and high-MAFB^+^ in both smoking and non-smoking patients and found that the survival was significantly lower in the high-MAFB^+^ group only in smokers (Figure 4A).

The effects of cigarette smoking on men and women have long been a subject of controversy [19]. The samples included female smoker *n* = 24, female non-smoker *n* = 40, male smoker *n* = 51, and male non-smoker *n* = 5. Our results showed that female smokers had significantly lower OS rates (Appendix A). In men, an accurate comparative analysis could not be performed as the number of non-smokers was only about 10% of smokers (Appendix A). For women, we separately compared the survival curves for MAFB expression intensity for non-smokers and smokers. The results showed no significant difference in survival by MAFB expression intensity in the non-smoker group, but a dramatic difference in OS and DFS was observed in the smoker group. Since most of the men were smokers, we could not obtain data on survival curves for non-smokers; however, smokers showed a significant difference in DFS according to the intensity of MAFB (Figure 4C). These data indicate that MAFB could be a prognostic TAM marker for patients with smoking habits with lung adenocarcinoma.

## 3. Discussion

TAMs and AMs are thought to coexist in the lung tissues from the early stages of cancer [22]. However, the cell-specific expression of MAFB remains largely unidentified. In our previous report, tumor samples from patients with lung adenocarcinoma showed MAFB expression in locations comparable to CD68- and CD204-positive TAMs and were abundant in severe stages of cancer [21]. In this study, the scRNA-seq analysis of patients with lung cancer showed MAFB expression in monocytes of tumor and advanced tumor tissue, while no other markers (CD204, CD68, CD206) were expressed (Figure 1J). In particular, in myeloid cluster 8, MAFB expression increased in tumors and advanced tumors. Similar expression patterns for CCR2 were observed in Appendix A. CCR2 is the receptor for CCL2 which induces monocyte infiltration in tumors including lung cancer [23]; therefore, MAFB may be a potential indicator for monocyte infiltration. Although it is difficult to measure the actual percentage of monocyte infiltration, the CCR2-expressing cells in this analyzed data accounted for approximately 3% of the total cells in normal lung tissue and increased to 6% or 11% in tumors or advanced tumors, respectively (Appendix A). This may indicate that TAM infiltration increases as the tumor stage advances, but it is difficult to clearly measure the extent of TAM infiltration considering conditions such as tumor removal site, sample preparation method, and individual differences. Further study is required to analyze whether MAFB can be established as a marker of invasion.

Moreover, unlike other TAM markers, MAFB was not expressed in AM. Our results were consistent with previous reports stating that Mafb was highly expressed in macrophage-colony stimulating factor (M-CSF)-derived macrophages but not expressed in the alveolar granulocyte macrophage-colony stimulating factor (GM-CSF)-derived macrophages. Since MAFB inhibits the self-renewal of macrophages, AMs have self-renewal ability [24]. Hence, our study suggests that MAFB shows a higher specificity to the macrophage/monocyte cell population than other cancer markers studied and could be used to identify patients with early stages of lung adenocarcinoma.

Previous studies have identified CD204^+^ TAMs as prognostic markers in non-small cell lung carcinoma (NSCLC), especially in lung adenocarcinoma [25], and the combined use of CD47 and CD68 was reported to predict the survival of eastern-Asian patients with NSCLC [26]. Further, CD68^+^CD163^+^ or CD68^+^CD206^+^ markers were used to identify M2-polarized TAMs in lung adenocarcinoma. The levels of M2 macrophages (CD68^+^CD206^+^) were positively associated with peritumoral lymphatic microvessel density, but negatively associated with the patient’s prognosis [27]. Moreover, the accumulation of CD163^+^ macrophages is closely correlated with a poor prognosis in lung cancer, and the increased density of CD68^+^CD163^+^ macrophages in tumor nests and stroma was associated with lymph node metastases [27]. However, no such association was observed with recurrence-free survival, OS, and TNM stages [14,28]. Similarly, CD68+CD163+M2 were also correlated with OS and DFS in NSCLC. A higher correlation was observed between increased infiltration of macrophages and clinical characteristics, including LUSC, EGFR status, and smoking habits [29]. However, the use of MAFB as a predictive marker for the survival of patients with non-metastatic lung adenocarcinoma remains unidentified. Our results showed that MAFB^+^ cell density correlated with clinicopathological characteristics in patients with stage I, II, and III lung adenocarcinoma. A higher MAFB^+^ cell density correlated with poor clinical outcomes, including poor pathologic stage, higher recurrence rate, nodal involvement, lymphatic permeation, and vessel invasion. An association with high hazards rate, poor OS, and DFS was also observed among these patients.

Smoking habits or sex differences did not show significant differences in their correlation with the OS and DFS in MAFB^+^ cells. However, one of the limitations of our study was the relatively small sample size of non-smoking males. Even though only Japanese patients were included in this study, our results were consistent with previous reports where the risk of lung cancer was comparable in both women and men exposed to tobacco smoke in patients from Germany and Italy [30]. Compared to squamous cells or small cell carcinomas, adenocarcinoma was reported to have a weak association with tobacco smoking in women from France [19]. Interestingly, smoking was significantly correlated with a higher density of CD68- and CD204-positive macrophages in tumor stroma [25,31]. It has also been reported that *Mafb* expression is upregulated in macrophages following exposure to cigarette smoke in a mouse model [10]. However, whether the increase in *MAFB* expression was observed in resident or infiltrating macrophages remained unclear. Our results indicate that *MAFB* was expressed in the monocyte-derived macrophages, but not tissue-resident macrophages, suggesting higher specificity of *MAFB* expression in TAMs than other markers, CD68 and CD204. Moreover, all patients with smoking habits showed higher MAFB^+^ cell density and were at risk of poor OS and DFS, suggesting that MAFB could be a prognostic TAM marker in smoking patients with early-stage lung adenocarcinomas. We suggest that the correlation between MAFB^+^ cells density with OS and DFS in smokers and/or non-smokers patients did not seem to be a sex-related relationship.

Our results suggest that MAFB^+^ cells could be a suitable predictor for severity and a prognostic marker for hazard rate, OS, and DFS in patients with non-metastatic lung adenocarcinoma. Moreover, we suggest that a higher MAFB^+^ cell density in patients with smoking habits could also be associated with poor overall and disease-free survival; however, this association is not sex-related.

## 4. Material and Methods

### 4.1. Single-Cell RNA Sequencing (scRNA-seq) Analysis

The single cell RNA-seq recently generated from 44 patients with treatment-naïve lung adenocarcinoma were analyzed (GEO database accession GSE131907). Single-cell RNA raw data included normal lung tissue (*n* = 11), tumor tissue (stage I and III, *n* = 7), and advanced tumor tissue (stage IV, *n* = 4).

The raw data was downloaded, and we used Scanpy (v1.7.2) for the following analyses. The initial cell number and gene number were 208,506 and 29,634, respectively. We extracted highly variable genes using “scanpy.pp.highly_variable_genes” function, and 2243 genes were extracted. We conducted dimension reduction with PCA using the “scanpy.tl.pca” function and UMAP using the “scanpy.pp.neighbors” function and the “scanpy.tl.umap” function. By clustering with the Leiden Method using the “sc.tl.leiden” function, cells were divided into 34 clusters and six clusters which contained myeloid cells (41,726 cells) were extracted. Dimension reduction and clustering were conducted on these extracted cells using the same method for all cells. For information about the parameters for these analyses, please refer to our GitHub pages (https://github.com/Teppei-Nishino/TAM, accessed on 30 August 2022)).

We clustered myeloid cell lineages, Alveolar Macrophages (AM), Dendritic cells (DC), Macrophage Lineage (ML), and Monocytes (Mo), and the analysis was performed using only these clusters. Statistical analysis and visualization were performed using functions from Scanpy.

### 4.2. Immunostaining of Human Cancer Tissues

We analyzed the cancerous tissues of patients with lung adenocarcinoma (*n* = 120) from the Tsukuba Human Tissue Biobank Center (THB) at the University of Tsukuba Hospital. Frozen human lung tumor tissues were sectioned (5 μm), stained, and visualized, as previously described [21], using 1:50 anti-MAFB (clone OTI2A6; Lifespans Biosciences, Seattle, WA, USA). Means of three positively stained field areas relative to the tissue area (MAFB^+^ cells/tissue area) were morphometrically quantified (Appendix A) using a BZ-X800 analyzer (Keyence, Itasca, IL, USA).

### 4.3. Evaluation of Clinicopathological Features

The clinical characteristics of patients with lung adenocarcinoma (*n* = 120) were retrieved from the clinical records of the University of Tsukuba. The following clinicopathological factors were considered: age (<70 years versus ≥70 years), sex (female vs. male), smoking history (non-smokers versus smokers), local metastasis to lymph nodes (nodal involvement; N0 versus N1), recurrence (positive vs. negative), pathological stage (I, II, and III), lymphatic permeation (present vs. absent), vascular invasion (present vs. absent), and pleural infiltration (present vs. absent). The UICC TNM staging system (The Union for International Cancer Control staging system for tumor size, lymphatic involvement, and metastasis) was used to classify the severity and extent of the cancer stage.

### 4.4. Statistical Analysis

Data are expressed as the mean ± SEM and analyzed using Welch’s *t*-test. The correlations between the grade of MAFB^+^ cells density and the clinicopathological factors were evaluated through Fisher’s exact test. The Kaplan–Meier method was used to estimate the overall survival time and the disease-free survival, while the difference in survival was compared using the log-rank test, the two paired groups using the Wilcoxon test, and the different survival distributions using the Tarone–Ware test. In survival analysis, we used Dunn–Šidák correction to adjust the *p*-value, and the curve comparisons were calculated using the Cox hazard test. The following variables were considered, MAFB^+^ cells density, sex, age, smoking, cancer recurrence, tumor stage, nodal involvement, lymphatic permeation, vessel invasion, and pleural infiltration. The correlation between MAFB^+^ cell density and overall survival (OS) or disease-free survival (DFS) was evaluated using the Pearson correlation test. Differences were considered statistically significant at *p* < 0.05.

## Figures and Tables

**Figure 1 ijms-23-09945-f001:**
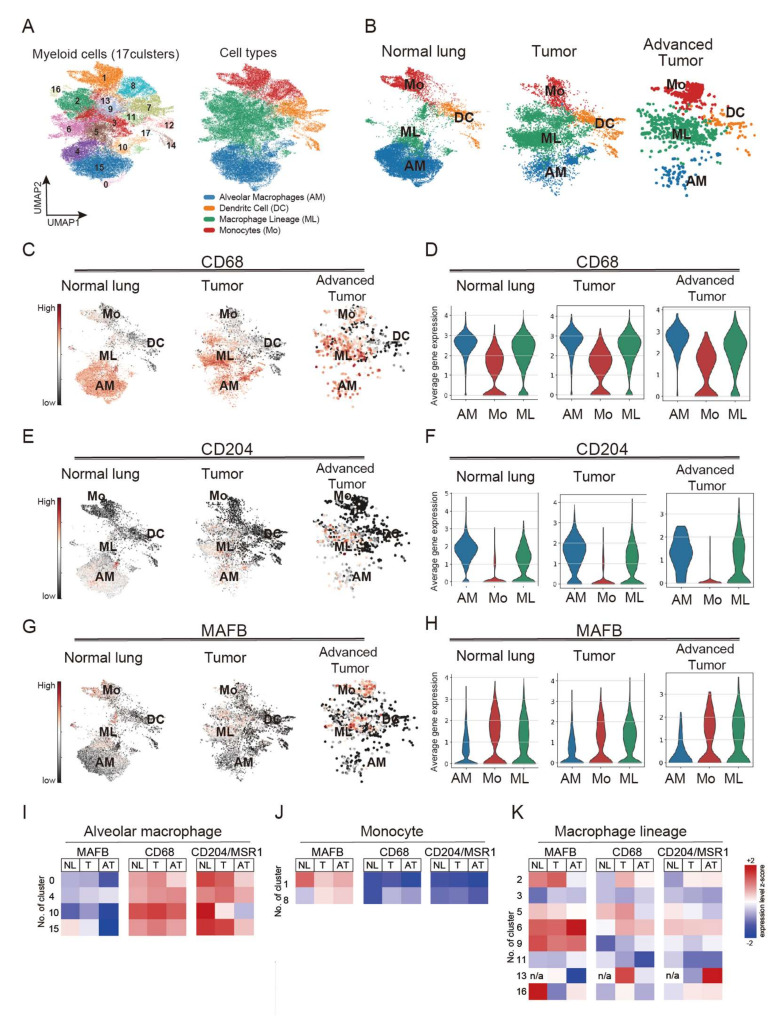
Single-cell RNA sequencing (scRNA-seq) analysis obtained from 44 patients with treatment-naive lung adenocarcinoma. Single-cell RNA raw data included normal lung tissue (*n* = 11), tumor tissue (stage I and III, *n* = 7), and advanced tumor tissue (stage IV, *n* = 4). Raw data were downloaded and processed using sctransform function in Seurat (v3). (**A**) Identified 17 clusters of the myeloid population. (**B**) monocytes (Mo), alveolar macrophages (AM), macrophages lineage (ML), and dendritic cells (DC) cluster distribution in normal lung tissues, tumors, and advanced tumors. (**C**,**D**) CD68 expression pattern in normal lung tissues, tumor, and advanced tumor. (**E**,**F**) CD204 expression pattern in normal lung tissues, tumor, and advanced tumor. (**G**,**H**) MAFB expression pattern in normal lung tissues, tumor, and advanced tumor. (**I**) Heatmap analysis of the expression of MAFB, CD68, and CD204 in AM. (**J**) Heatmap analysis of the expression of MAFB, CD68, and CD204 in monocytes. (**K**) Heatmap analysis of the expression of MAFB, CD68, and CD204 in macrophage lineage.

**Figure 2 ijms-23-09945-f002:**
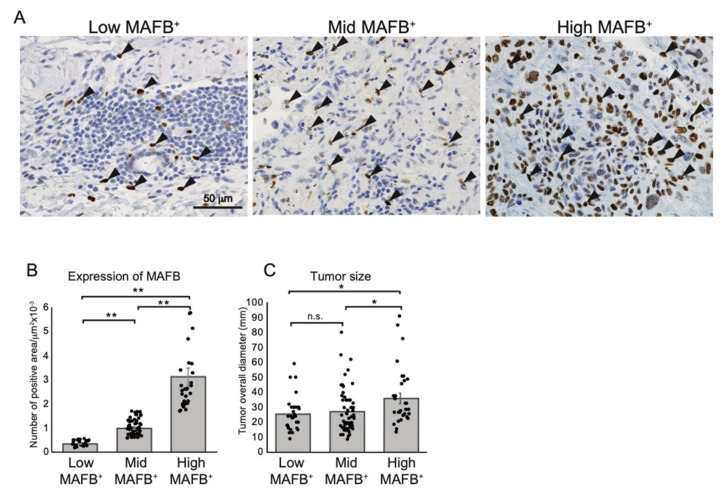
Grouping non-metastatic lung adenocarcinoma tissue according to MAFB^+^ cells density. (**A**) Representative data of immunohistochemical analysis of human lung adenocarcinomas with anti-human MAFB. Arrows point out the MAFB-positive cells. (**B**) MAFB-positive area relative to tissue area (MAFB/tissue area) was morphometrically quantified. Tissue samples were grouped into top 25% (high-MAFB^+^ group, MAFB expression area/tissue area = 0–0.005 (*n* = 30)), 25–50% (mid-MAFB^+^ group, MAFB expression area/tissue area = 0.006–0.016 (*n* = 59)), and bottom 25% (low-MAFB^+^ group, MAFB expression area/tissue area = 0.017–0.121 (*n* = 31)). (**C**) MAFB expression in three groups was tested for correlation to tumor sizes. Data are presented as means ± SEM; data is considered significant at * *p* < 0.05, ** *p*< 0.01.

**Figure 3 ijms-23-09945-f003:**
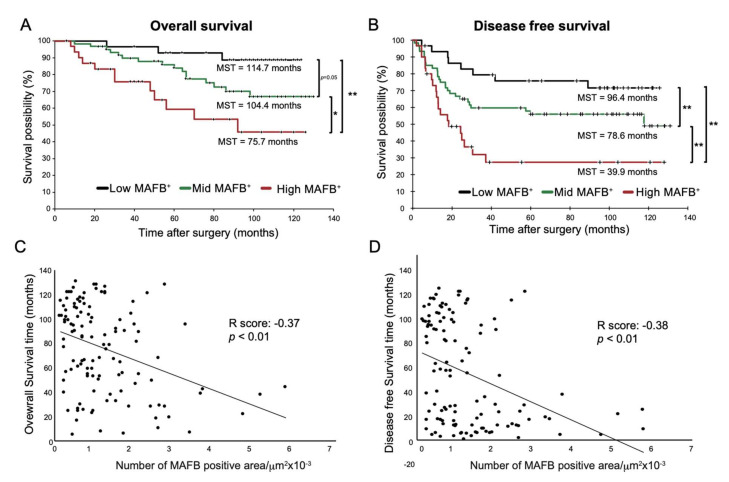
OS and DFS of low-, mid-, and high- in MAFB^+^ cells Kaplan–Meier analysis of (**A**) overall survival and (**B**) disease-free survival of the three groups: low-MAFB^+^, mid-MAFB^+^, and high-MAFB^+^. Difference in survival was compared using log-rank test. Pearson correlation analysis was performed between MAFB expression and (**C**) survival time (R score: −0.366, *p* = 0.000043) and (**D**) disease-free survival (R score: −0.378, *p* = 0.000023). Data are presented as means ± SEM; data is considered significant at * *p* < 0.05; ** *p* < 0.01.

**Figure 4 ijms-23-09945-f004:**
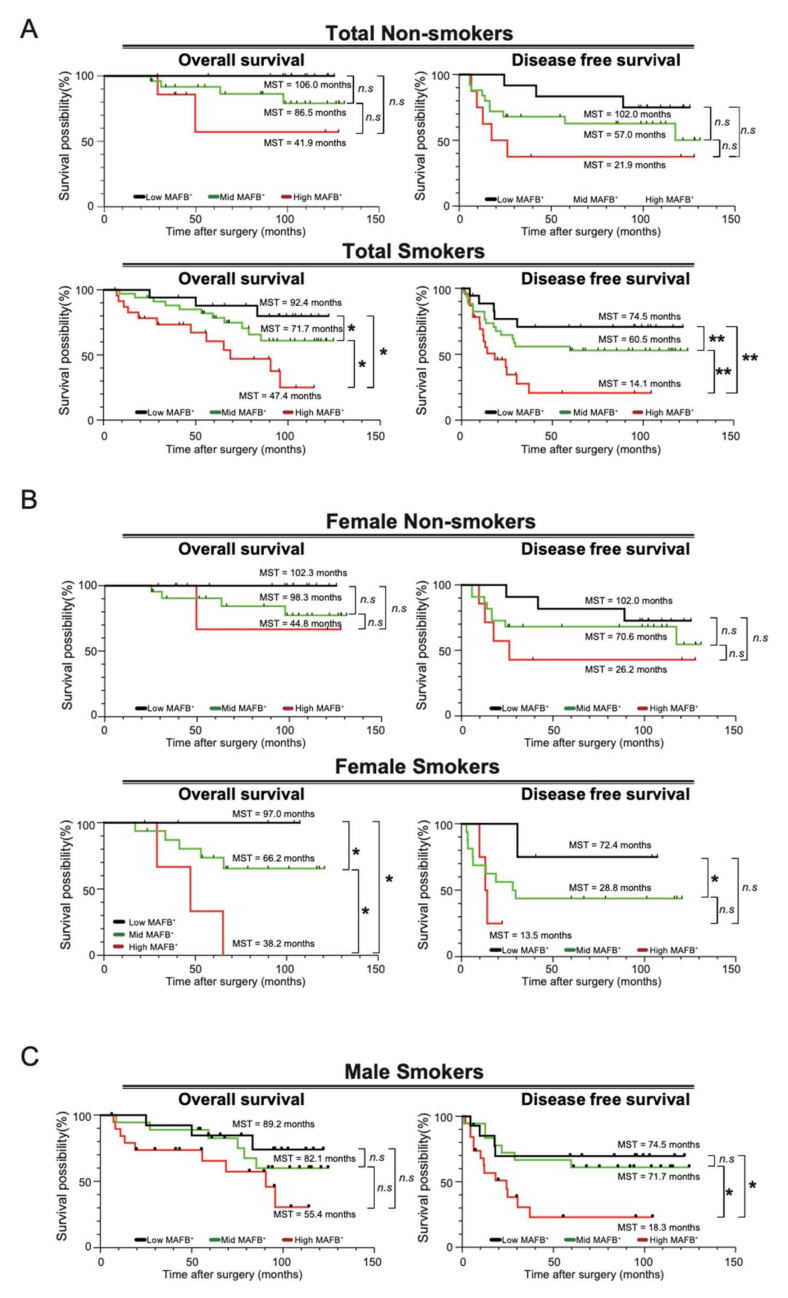
OS and DFS analysis in smokers and non-smoker patients. Kaplan–Meier analysis of (**A**) the overall survival and disease-free survival of the low-MAFB^+^, mid-MAFB^+^, and high-MAFB^+^ in the total smoking and non-smoking patients; (**B**) the overall survival and disease-free survival of the low-MAFB^+^, mid-MAFB^+^, and high-MAFB^+^ in female smoker and non-smoker groups; and (**C**) the overall survival and disease-free survival of the low-MAFB^+^, mid-MAFB^+^, and high-MAFB^+^ in male smoker and non-smoker groups. Difference in survival was compared using log-rank test. *, *p* < 0.05; **, *p* < 0.01.

**Table 1 ijms-23-09945-t001:** Correlation between MAFB^+^ cell density and the clinicopathological factors in non-metastatic lung adenocarcinoma.

	Low-MAFB^+^ (≤0.005)	Mid-MAFB^+^ (0.006–0.016)	High-MAFB^+^ (≥0.017)	*p-Value of* *Fisher’s Exact Test*
Variables	No of Case *n* = 30 (25%)	No of Case *n* = 59 (49%)	No of Case *n* = 31 (26%)	
**Age (y)**				
<70	20 (67%)	39 (66%)	15 (48%)	0.2249
≥70	10 (33%)	20 (34%)	16 (52%)	
**Gender**				
Male	15 (50%)	21 (36%) *	20 (65%) *	**0.0302**
Female	15 (50%)	38 (64%) *	11 (35%) *	
**Smoking history**				
Never	12 (40%)	25 (42%)	8 (26%)	0.3046
Former or current	18 (60%)	34 (58%)	23 (74%)	
**Clinical Stage**				
I (*n* = 57)	24 (42%) ***	30 (53%)	3 (5%) ***	**<0.001**
II + III (*n* = 63)	6 (10%) ***	29 (46%)	28 (44%) ***	
**Cancer Recurrence**				
Negative	22 (73%) *	33 (56%)	10 (32%) *	**0.006**
Positive	8 (27%) *	26 (44%)	21 (68%) *	
**Nodal involvement**				
Negative (N−)	24 (80%) ***	32 (54%)	4 (13%) ***	**<0.0001**
Positive (N+)	6 (20%) ***	27 (46%)	27 (87%) ***	
**Lymphatic permeation**				
Ly(–)	26 (87%) ***	34 (58%)	12 (39%) **	**<0.001**
Ly(+)	4 (13%) ***	25 (42%)	19 (61%) **	
**Vessel invasion**				
V(–)	23 (77%) ***	26 (44%)	5 (16%) ***	**<0.0001**
V(+)	7 (23%) ***	33 (56%)	26 (84%) ***	
**Pleural infiltration**				
PL(–)	23 (77%) *	31 (53%)	13 (42%)	**0.019**
PL(+)	7 (23%) *	28 (47%)	18 (58%)	

According to density of cells expressing MAFB, 120 lung adenocarcinoma patients with stages I, II, and III were grouped into low-MAFB^+^, mid-MAFB^+^, and high-MAFB^+^ cell density groups. Correlation between MAFB expression and clinical factors among groups was recorded and statistically analyzed using Fisher’s exact test, * *p* < 0.05, ** *p* < 0.001, *** *p* < 0.0001.

**Table 2 ijms-23-09945-t002:** Univariate analysis of disease-free survival and overall survival in non-metastatic lung adenocarcinoma stages.

	Univariate Analysis				Multivariate Analysis			
	Disease-Free Survival		Overall Survival		Disease-Free Survival		Overall Survival	
	Hr (95% Ci)	*p* Value	Hr (95% Ci)	*p* Value	Hr (95% Ci)	*p* Value	Hr (95% Ci)	*p* Value
MAFB (low versus mid)	2.998 (0.9882–12.95)	0.0828	3.039 (1.001–13.13)	0.079	1.665 (0.5008–7.565)	0.4468	1.966 (0.5985–8.859)	0.3086
MAFB (low versus high)	7.423 (2.393–32.44)	**0.0018**	8.105 (2.620–35.36)	**0.0011**	1.773 (0.4873–8.669)	0.4230	2.001 (0.5587–9.646)	0.3263
Gender (male versus female)	1.856 (0.9236–3.847)	0.0861	1.826 (0.9081–3.790)	0.095				
Smoking (Yes versus No)	3.115 (1.369–8.376)	**0.0122**	0.3208 3.117 (1.368–8.390)	**0.0123**	3.106 (1290–8.724)	**0.0182**	2.989 (1.231–8.437)	**0.0235**
Age (<70 versus ≥70)	1.010 (0.9697–1.058)	0.6397	1.013 (0.9715–1.062)	0.5578				
Pstage (I versus II + III)	7.500 (3.194–20.72)	**<0.0001**	8.464 (3.614–23.24)	**<0.0001**				
Nodal involvement (Yes versus No)	6.497 (2.880–16.71)	**<0.0001**	7.723 (3.411–19.85)	**<0.0001**	4.382 (1.555–13.88)	**0.0078**	5.304 (1.845–17.12)	**0.0032**
Lymphatic invasion (Yes versus No)	3.072 (1.522–6.398)	**0.002**	3.294 (1.626–6.887)	**0.0011**	1.049 (0.4812–2.384)	0.9064	1.094 (0.4918–2.522)	0.8285
Vessel invasion (Yes versus No)	5.444 (2.377–14.71)	**0.0002**	4.875 (2.136–13.14)	**0.0005**	1.843 (0.6468–5.939)	0.2753	1.370 (0.4730–4.458)	0.5786
Pleural infiltration (Yes versus No)	2.740 (1.350–5.803)	**0.0062**	2.644 (1.306–5.585)	**0.008**	1.228 (0.5665–2.793)	0.6108	1.216 (0.5498–2.805)	0.6363

Statistically significant differences between groups were determined using Cox proportional hazard model (*p* < 0.05). HR, hazard ratio; CI, confidence interval.

## Data Availability

The datasets used and/or analyzed during the current study are available from the corresponding author upon reasonable request. All source code is available from our GitHub page (https://github.com/Teppei-Nishino/TAM, accessed on 30 August 2022)).

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
