# Peer review of "Transcription Factor MAFB as a Prognostic Biomarker for the Lung Adenocarcinoma"

_ijms, 2022, doi:10.3390/ijms23179945_

Round 1

Reviewer 1 Report

Well written article with interesting aspects. The validity could be improved by analyzing further patient samples. Is there evidence that MAFB expression changes over the course of the disease?

Author Response

Reviewer 1:  Well written article with interesting aspects. The validity could be improved by analyzing further patient samples. Is there evidence that MAFB expression changes over the course of the disease?

Response:  Thank you for your question. In our previous published paper (Yadav et al., Biochem. Biophys. Res. Commun., 521, no. 3, pp. 590–595, 2020) the immunohistochemical analysis indicated that the number of MAFB-positive cells in tissue samples of the patients with severe lung adenocarcinoma was more than twofold higher than in samples from patients with moderate lung adenocarcinoma. These data indicate that MAFB expression may increase over the course of the disease. However, it is difficult to check MAFB expression in the same patient. To answer the reviewer’s question appropriately, further analysis using a sample from an early to an advanced stage in the same patient is necessary.

Reviewer 2 Report

The authors of the article "Transcription factor MAFB as a prognostic biomarker for lung adenocarcinoma" did an excellent job of finding MAFB as a prognosis marker in the metastasis of the tumor. They basically demonstrate that MAFB is expressed in TAMs and that this expression corresponds with lymph node metastasis. The authors have performed admirably and utilized appropriate statistical analysis techniques in their study. I have no other comments on the study, except that the titles for the results should be made more informative and self-explanatory.

For example, under result heading 1 "MAFB is not expressed in alveolar macrophages in both normal and cancerous tissue," the author can emphasize the positive rather than the negative outcome.

In addition, the author should discuss the percentage of macrophage infiltration in general as well as their personal findings.

Author Response

Reviewer 2: The authors of the article "Transcription factor MAFB as a prognostic biomarker for lung adenocarcinoma" did an excellent job of finding MAFB as a prognosis marker in the metastasis of the tumor. They basically demonstrate that MAFB is expressed in TAMs and that this expression corresponds with lymph node metastasis. The authors have performed admirably and utilized appropriate statistical analysis techniques in their study. I have no other comments on the study, except that the titles for the results should be made more informative and self-explanatory.

Comment 1: For example, under result heading 1 "MAFB is not expressed in alveolar macrophages in both normal and cancerous tissue," the author can emphasize the positive rather than the negative outcome.

Response: Thank you for your critical suggestion, we modified the subtitle of the Result section according to the reviewer’s comment. Thanks to this suggestion, we believe that the revised subtitle in the manuscript is clearer. Other changes were made in line with this comment as follows:

3.1“MAFB is not expressed in alveolar macrophages in both normal and cancerous tissue”

-> MAFB is specifically expressed in monocyte/macrophages but not alveolar macrophages in both normal and cancerous tissue 

3.2” Tissues from stages I, II, and III lung adenocarcinoma grouped according to the MAFB+ cells density”

-> higher MAFB+ cell density may be associated with poor clinical prognosis among lung cancer patients

3.3. MAFB+ cells density and survival curve

-> high-MAFB+ cell density indicated a higher mortality risk

3.4. Relation between MAFB expression and survival of the smoker

-> MAFB could be a prognostic TAM marker for patients with smoking habits with lung adenocarcinoma

Comment 2: In addition, the author should discuss the percentage of macrophage infiltration in general as well as their personal findings.

Response: Thank you for pointing this out. In general, it is difficult to determine the exact percentage of infiltrated macrophages and monocytes in a tumor. This is because it depends on the location and size of the tumor sample and the individuality of the patient. Previous reports have shown that CCL2/CCR2 signaling is important for the infiltration of lung cancer tissue by monocytes (Schmall A, et al., Am. J. Respir. Crit. Care Med. 191(4):437–47 2015). Therefore, we checked the number of CCR2-positive monocytes/macrophages in the data used in this study. We found that the number of CCR2-positive cells was about 3% in normal lungs, while it increased to 6% and 11% in the Tumor and Advanced Tumor samples, respectively. These results have been added to the new Supplementary Figure. S3B. MAFB is also expressed in about 60% of CCR2-positive cells, and MAFB may also be a marker for infiltrating cells.

Future studies should focus on MAFB-positive and CCR2-positive monocytes and deeply analyze whether the infiltration of these monocytes is an indicator of cancer grade. We have added a passage detailing this to the Discussion section (Line 316–328).